# Prevalence and mechanisms of evolutionary contingency in human influenza H3N2 neuraminidase

Ruipeng Lei [1], Timothy J. C. Tan [2], Andrea Hernandez Garcia[1], Yiquan Wang [1], Meghan Diefenbacher[3], Chuyun Teo[1], Gopika Gopan[4], Zahra Tavakoli Dargani[1], Qi Wen Teo[1,5], Claire S. Graham[1], Christopher B. Brooke [3,5], Satish K. Nair [1,2,5] & Nicholas C. Wu [1,2,5,6] ✉

Neuraminidase (NA) of human influenza H3N2 virus has evolved rapidly and been accumulating mutations for more than half-century. However, biophysical constraints that govern the evolutionary trajectories of NA remain largely elusive. Here, we show that among 70 natural mutations that are present in the NA of a recent human H3N2 strain, >10% are deleterious for an ancestral strain. By mapping the permissive mutations using combinatorial mutagenesis and next-generation sequencing, an extensive epistatic network is revealed. Biophysical and structural analyses further demonstrate that certain epistatic interactions can be explained by non-additive stability effect, which in turn modulates membrane trafficking and enzymatic activity of NA. Additionally, our results suggest that other biophysical mechanisms also contribute to epistasis in NA evolution. Overall, these findings not only provide mechanistic insights into the evolution of human influenza NA and elucidate its sequence-structure-function relationship, but also have important implications for the development of next-generation influenza vaccines.

Influenza A virus, which causes around 300,000 to 650,000 deaths yearly over the world[1], continues to be a major global health concern. Influenza A virus has two surface glycoproteins, hemagglutinin (HA) and neuraminidase (NA), that act as the major antigens. Because of the need to constantly escape from herd immunity, both HA and NA of human influenza A virus evolve rapidly. Most studies on the evolution of human influenza A virus focus on HA, due to its dominant role in vaccine development[2]. Nevertheless, recent studies have shown that NA immunity also contributes to protection against influenza infection[3–6]. These findings suggest that NA can be an effective vaccine target[7]. However, there is a lack of knowledge about the evolutionary biology of NA.

NA is a type II transmembrane protein that forms a homotetramer. The natural function of NA is to cleave the sialic acids from cellular receptors to facilitate virus release and from newly synthesized HA to prevent virion aggregation[8]. After entering the human population in 1968, H3N2 has accumulated around 75 amino acid mutations in NA, which account for 16% of the entire protein[9]. Such an extensive evolution of NA can at least be partly attributed to epistasis, where the fitness of a mutation depends on the genetic background. For example, our recent study indicates that the emergence of mutations in an antigenic region of human H3N2 NA is contingent on the evolution of other parts of the sequence[9]. Similarly, the emergence of oseltamivir resistance in seasonal H1N1 virus is contingent on permissive

[1]Department of Biochemistry, University of Illinois at Urbana-Champaign, Urbana, IL 61801, USA. [2]Center for Biophysics and Quantitative Biology, University of Illinois at Urbana-Champaign, Urbana, IL 61801, USA. [3]Department of Microbiology, University of Illinois at Urbana-Champaign, Urbana, IL 61801, USA. [4]Department of Chemistry, University of Illinois at Urbana-Champaign, Urbana, IL 61801, USA. [5]Carl R. Woese Institute for Genomic Biology, University of Illinois at Urbana-Champaign, Urbana, IL 61801, USA. [6]Carle Illinois College of Medicine, University of Illinois at Urbana-Champaign, Urbana, IL 61801, USA. ✉e-mail: nicwu@illinois.edu

mutations[10–12]. Although epistasis is central to the understanding of evolutionary trajectories[13,14], the prevalence of epistasis in the evolution of human influenza NA remains elusive.

Mechanisms of epistasis can be categorized into two classes, specific and non-specific[15]. Specific epistasis, which describes a nonlinear mapping from sequence to physical property, typically involves mutations that are spatially proximal on the protein structure and physically interacting with each other. In contrast, non-specific epistasis, which describes a nonlinear mapping from physical property to phenotype, depends less on the spatial distance between mutations. Both types of epistasis are involved in the natural evolution of human influenza virus. For example, specific epistasis is observed in a network of HA mutations that coordinately modulate the evolution of receptor-binding mode[16,17]. On the other hand, non-specific epistasis is observed in the evolution of influenza nucleoprotein (NP), in which the emergence of destabilizing mutations is contingent on the presence of stabilizing mutations to maintain a melting temperature of at least 43 °C for optimum transcriptional activity[18]. As a result, characterization of epistasis can provide mechanistic insights into the biophysical constraints that govern the evolutionary trajectories of influenza proteins.

In this study, we systematically identified natural mutations in human H3N2 NA that were unfit in an ancestral strain but subsequently emerge. We further mapped an epistatic network among natural mutations in human H3N2 NA using combinatorial mutagenesis and next-generation sequencing. Microscopy and flow cytometry analyses showed that membrane trafficking efficiency and enzymatic activity contribute to epistasis between NA mutations. Using biophysical assays and X-ray crystallography, we also demonstrated that non-additive stability effect led to specific epistasis in NA evolution. Overall, this study represents an in-depth analysis of epistasis in the natural evolution of human H3N2 NA as well as the underlying mechanisms.

## Results

### Prevalence of evolutionary contingency in human H3N2 NA

Positive epistasis can lead to evolutionary contingency, where a deleterious mutation in an ancestral strain becomes neutral and emerges after one or more permissive mutations arise[15]. To probe the prevalence of evolutionary contingency in human H3N2 NA, we constructed 70 single mutants of A/Hong Kong/1968 (HK68) NA, each representing an amino-acid mutation from HK68 to A/Victoria/361/2011 (Vic11) (Supplementary Fig. 1a). The fitness effects of these HK68 NA mutants were then measured by a virus rescue experiment (Fig. 1a). Eight out of the 70 (11%) mutants had at least two-log drop in titer compared to the wild type (WT), indicating the presence of evolutionary contingency. Except H155Y, which located at the protomer interface, the other seven mutations were all far away from the protomer interface and were solvent exposed (Fig. 1b). For our downstream experiments, we focused on two mutations with the most extreme phenotypes, namely N336H and N387K, both of which did not yield any detectable titer in the virus rescue experiment. These mutational fitness effects were unlikely due to the difference in HA, as replacing HK68 HA with Vic11 HA did not affect the WT virus titer (Supplementary Fig. 2a).

### Natural emergence of compatibility with N336H and N387K

To narrow down the time period when N336H and N387K became neutral to the virus, we introduced N336H and N387K into the NAs of other H3N2 strains from different years, including A/Bilthoven/17938/1969 (Bil69), A/Bilthoven/21438/1971 (Bil71), A/Albany/1/1976 (Alb76), A/Bangkok/1/1979 (BK79), A/Beijing/353/1989 (Bei89), A/Shandong/9/

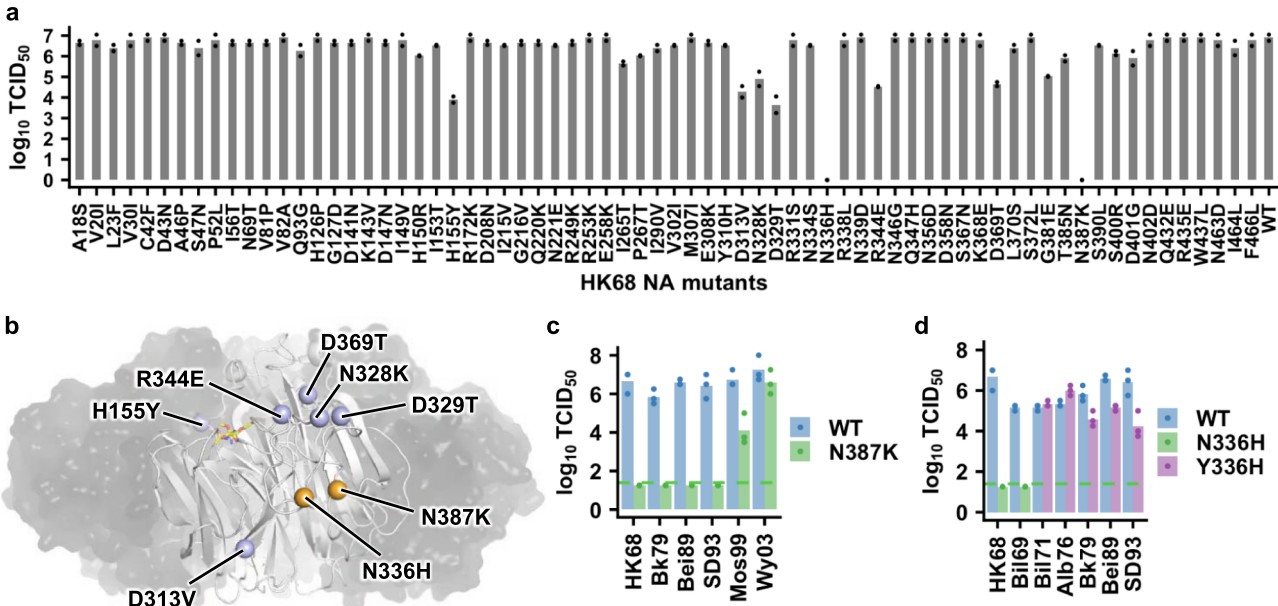

**Fig. 1 | Virus rescue experiments of natural mutations. a** The amino acid sequences of HK68 NA and Vic11 NA differed by 70 amino acid substitutions. These 70 mutations were individually introduced into HK68 NA. Their replication fitness was examined by a virus rescue experiment. Virus titer was measured by 50% cell culture infectious dose (TCID$_{50}$). Each bar represents the mean of two independent biological replicates. Each datapoint represents one biological replicate. **b** The locations of the eight mutations that showed at least two-log decrease in virus titer compared to the wild type (WT) are shown as spheres on one protomer of NA that is in white cartoon representation, while the other three protomers are shown as semitransparent black surface (PDB 3TIA)[21]. Sialic acid in the active site is shown as yellow sticks. Locations of the two mutations that did not yield any detectable titer, namely N336H and N387K, are in orange. The other six mutations are in blue. **c, d** Fitness effects of **c** N387K and **d** N/Y336H on different strains were examined by a virus rescue experiment. While the WTs of HK68 and Bil69 had an Asn at residue 336, the WTs of Bil71, Alb76, Bk79, Bei89, and SD93 had a Tyr at residue 336. Virus titer was measured by TCID$_{50}$. Each bar represents the mean of three independent biological replicates. Each datapoint represents one biological replicate. The green dashed line represents the lower detection limit. Source data are provided as a Source data file.

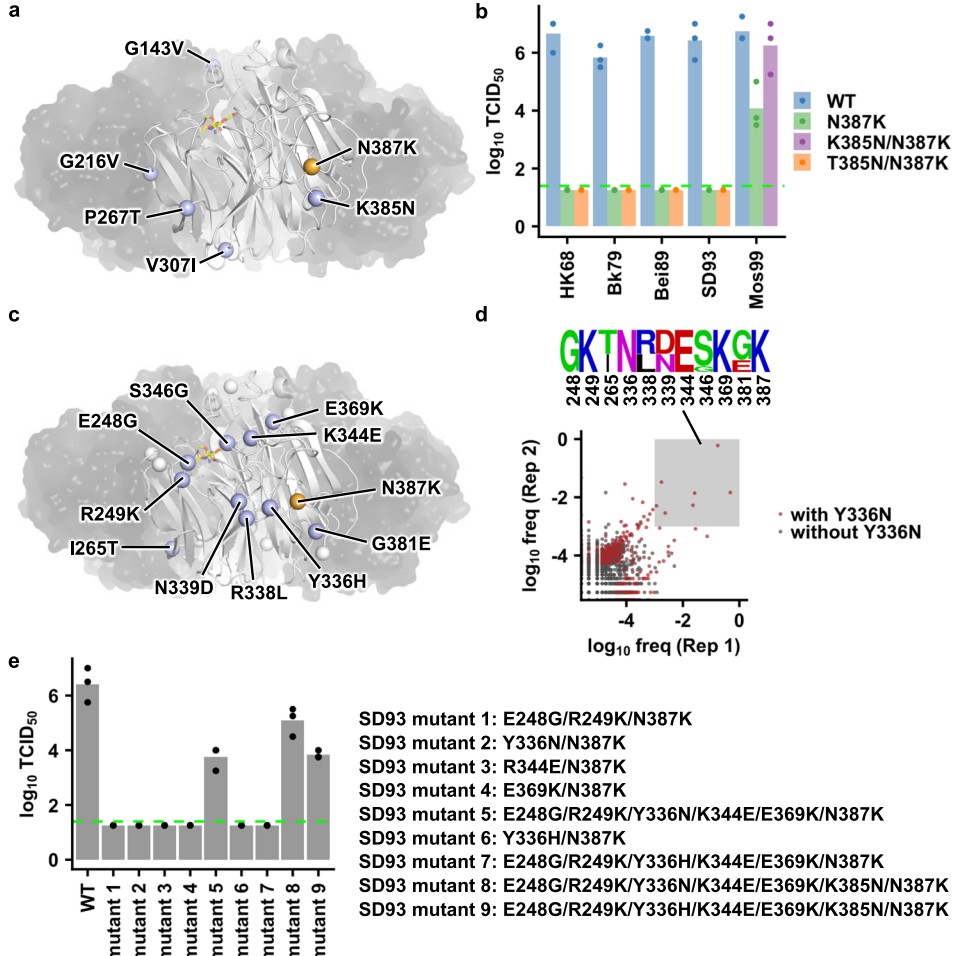

**Fig. 2 | Permissive mutations for NA N387K in Mos99 and SD93. a** Locations of natural mutations from Mos99 to Wy03 are shown as blue spheres on one protomer of NA that is in white cartoon representation, while the other three protomers are shown as semitransparent black surface (PDB 3TIA)[21]. Location of N387K is shown as an orange sphere. Sialic acid in the active site is shown as yellow sticks. **b** The compensatory effects of mutating residue 385 to Asn in different chimeric H3N2 strains were examined by a virus rescue experiment. **c** Locations of natural mutations from SD93 to Mos99 are shown as blue or white spheres on one protomer of NA that is in white cartoon representation, while the other three protomers are shown as semitransparent black surface (PDB 3TIA)[21]. Blue spheres represent the mutations that were included in the SD93 combinatorial mutant library. **d** High-throughput identification of permissive mutations for N387K in SD93. The occurrence frequency of each variant in the post-passaged mutant library is shown. A de novo mutation Y336N was emerged during passaging. Datapoints that represent mutants with Y336N are in red. The amino acid sequences of variants with >0.1% frequencies in the post-passaged mutant libraries of both biological replicates are represented by a sequence logo. **e** The replication fitness of different SD93 mutants was examined by a virus rescue experiments. **b**, **e** Each bar represents the mean of three independent biological replicates. Each datapoint represents one biological replicate. Virus titer was measured by TCID₅₀. The green dashed line represents the lower detection limit. Source data are provided as a Source Data file.

1993 (SD93), A/Moscow/10/1999 (Mos99), and A/Wyoming/3/2003 (Wy03). While N387K was neutral in Wy03, it attenuated Mos99 and reduced the virus titer of Bk79, Bei89, and SD93 to below the detection limit (Fig. 1c). This observation indicates that the permissive mutations for N387K emerged between 1993 and 2003. Similarly, our result implies that the permissive mutations for N336H emerged between 1969 and 1971, since mutating residue 336 to His was neutral in Bil71 and Alb76 but reduced the virus titer of Bil69 to below the detection limit (Fig. 1d). Consistently, mutating NA residue 336 to His was neutral in the authentic H3N2 A/Udorn/1972, whereas N387K did not yield any detectable titer (Supplementary Fig. 2b).

## K385N is a permissive mutation for N387K
Next, we aimed to identify the permissive mutations for N387K. Since N387K decreased the titer of Mos99 by around two-log but was neutral in Wy03, we postulated that the difference in the NA sequences between Mos99 and Wy03 should include at least one permissive mutation for N387K. The NA head domains of Mos99 and Wy03 differed by five mutations (Fig. 2a and Supplementary Fig. 1b). One of

these mutations, K385N (Lys in Mos99 and Asn in Wy03), was adjacent to N387K. In fact, the double mutant K385N/N387K in Mos99 yielded a WT-like titer (Fig. 2b), demonstrating that K385N was a permissive mutation for N387K. Of note, K385N alone was neutral to the Mos99 (Supplementary Fig. 3a). Nevertheless, mutating residue 385 to Asn could not restore the fitness of N387K in earlier strains, including HK68, Bk79, Bei89, and SD93 (Fig. 2b), suggesting that additional permissive mutations were required for the natural emergence of N387K.

## Mapping additional permissive mutations for N387K
Although N387K decreased the fitness of Mos99, it reduced the titer of SD93 to below the detection limit (Fig. 1c). As a result, the difference in the NA sequences between SD93 and Mos99 should include additional permissive mutations for N387K. The NA head domains of SD93 and Mos99 differed by 18 mutations (Supplementary Fig. 1c). Since multiple mutations may be required simultaneously to restore the fitness of N387K in SD93, we decided to use a high-throughput approach that coupled combinatorial mutagenesis and

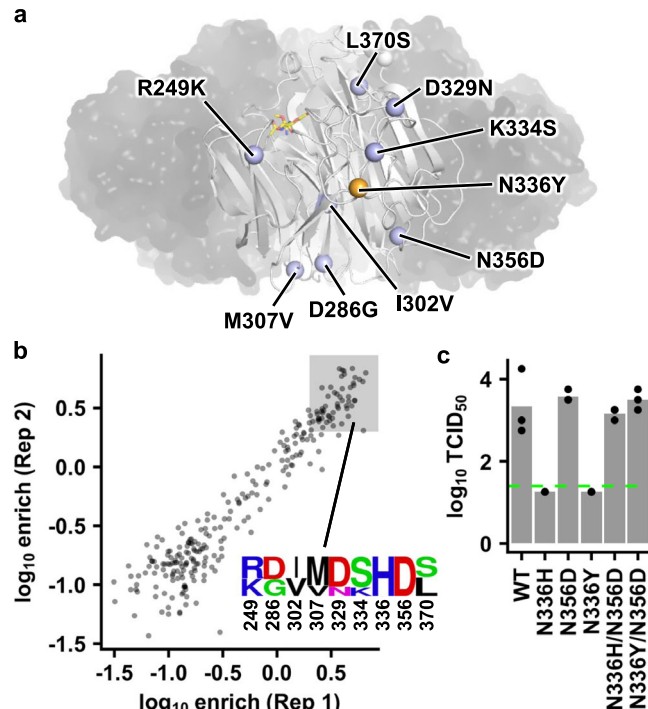

**Fig. 3 | Permissive mutations for NA N336Y in Bil69. a** Locations of natural substitutions from Bil69 to Bil71 are shown as blue spheres on one protomer of NA that is in white cartoon representation, while the other three protomers are shown as semitransparent black surface (PDB 3TIA)[21]. The location of N336Y is shown as an orange sphere. Sialic acid in the active site is shown as yellow sticks. **b** High-throughput identification of permissive mutations for N336Y in Bil69. The frequency enrichment of each variant in the post-passaged mutant library is shown. The amino acid sequences of variants with at least 2-fold enrichment ($\log_{10}$ enrichment ≥0.3) in the post-passaged mutant libraries of both biological replicates are represented by a sequence logo. **c** The replication fitness of different Bil69 mutants was examined by a virus rescue experiments. Each bar represents the mean of three independent biological replicates. Each datapoint represents one biological replicate. Virus titer was measured by $TCID_{50}$. The green dashed line represents the lower detection limit. Source data are provided as a Source data file.

next-generation sequencing[9,16,17] to identify permissive mutations for N387K. Briefly, we constructed a mutant library for SD93 that contained all possible combinations of 10 mutations ($n = 2^{10} = 1024$ variants), namely E248G, R249K, I265T, Y336H, R338L, N339D, K344E, S346G, E369K, and G381E (Fig. 2c), in the genetic background of N387K. Those 10 mutations were selected based on their distances to N387K in the protein structure and the primary sequence. The viral mutant library was then rescued and passaged once in MDCK-SIAT1 cells, which could minimize the emergence of cell-adaptive mutation[19].

Mutations E248G, R249K, Y336N, K344E, and E369K were enriched among variants with a frequency of >0.1% in the post-passaged mutant libraries of both biological replicates (Fig. 2d), suggesting that they could restore the fitness of N387K in SD93. Of note, Y336N was not included in the construction of the mutant library, and hence was a de novo mutation that emerged during viral passaging. Interestingly, Y336N represented a reversion to the ancestral stains, including HK68 and Bil69 (Supplementary Figs. 1a and 1d). While the fitness of each mutant in a mutant library could typically be estimated from its frequency change between the input mutant library and the post-passaged mutant library (i.e., frequency enrichment)[16], the absence of Y336N in the input mutant library prevented the use of frequency enrichment for fitness measurement.

To validate the findings from our high-throughput screening, we performed a series of virus rescue experiments with individually

constructed SD93 mutants (Fig. 2e). The pentamutant E248G/R249K/Y336N/K344E/E369K was indeed able to partially restore the virus titer of N387K. The virus titer of E248G/R249K/Y336N/K344E/E369K/N387K could be further increased by the addition of K385N. Of note, in the absence of N387K, these mutation combinations had a higher virus replication fitness than the WT (Supplementary Fig. 3b). In addition, there seemed to be a synergistic effect among these permissive mutations because none of E248G/R249K, Y336N, K344E, E369K, and K385N alone could restore the virus titer of N387K to a detectable level (Fig. 2b and e). As mentioned above, Y336N represented a reversion in the natural evolution of human H3N2 NA (e.g., reverting from Bil71 to Bil69, Supplementary Fig. 1d). In contrast, Y336H was the forward mutation in the natural evolution of human H3N2 NA (e.g., mutating from SD93 to Mos99, Supplementary Fig. 1c). Since both Y336N and Y336H enabled SD93 to mutate away from Tyr at residue 336, we also tested whether Y336H could act as a permissive mutation for N387K. However, both Y336H alone and the pentamutant E248G/R249K/Y336N/K344E/E369K could not restore the fitness of N387K (Fig. 2e). Nonetheless, although K385N could not restore the virus titer of N387K in the background of WT SD93 (Fig. 2b), it could partially do so in the background of E248G/R249K/Y336N/K344E/E369K (see mutant 9 in Fig. 2e). This observation shows that the combination of E248G/R249K/Y336N/K344E/E369K potentiates the ability of K385N to compensate the fitness defect of N387K. Together, our results demonstrate the complexity of epistatic networks that led to the natural emergence of N387K in human H3N2 NA.

### N356D is a permissive mutation for N336H

Besides N387K, N336H was another highly deleterious mutation in earlier strains (Fig. 1d), and yet emerged during the evolution of human H3N2 NA. Mutations between the NA head domains of Bil69 and Bil71 were identified as candidate permissive mutations for N336H (Supplementary Fig. 1d), since these two strains had drastically different compatibility with His336 (Fig. 1d). We focused on eight mutations that were spatially proximal to residue 336, namely R249K, D286G, I302V, M307V, D329N, K334S, N356D, and L370S (Fig. 3a). To identify which of these eight mutations were permissive mutations for N336H, we used the same high-throughput approach that was described above. Our Bil69 mutant library contained all possible combinations of the eight mutations of interest ($n = 2^8 = 256$ variants), in the background of N336H. Among variants that were enriched by at least 2-fold in the post-passaged mutant libraries of both biological replicates, N356D was highly conserved (Fig. 3b). Our virus rescue experiment further validated that N356D was a permissive mutation for N336H and was neutral when being introduced alone (Fig. 3c). While N336H represented a mutation from HK68 to Vic11, Bil71 had a Tyr at residue 336 (Supplementary Fig. 1d). Same as N336H, N336Y also reduced the titer of Bil69 to below the detection limit and the addition of N356D could restore it to the WT level (Fig. 3c). These results show that N356D is a permissive mutation for the natural evolution of residue 336, which in turn participates in a more extensive epistatic network that involves N387K (Fig. 2d and e).

### Enzymatic activities of H3N2 NA mutants

To understand the mechanisms of epistasis that involved N336H/Y and N387K, we first measured the NA activity on the surface of cells that were transiently transfected with different NA mutants. N387K drastically reduced the cell surface NA activity of Mos99 and SD93 (Fig. 4a, b), whereas both N336H and N336Y abolished the cell surface NA activity of Bil69 (Fig. 4c). The reduced cell surface NA activity of N387K in Mos99 and SD93 could be partially restored by K385N and E248G/R249K/Y336N/K344E/E369K, respectively (Fig. 4a, b). Similarly, the reduced cell surface NA activity of N336H and N336Y in Bil69 could be fully restored by N356D (Fig. 4c). Therefore, the cell surface NA activity

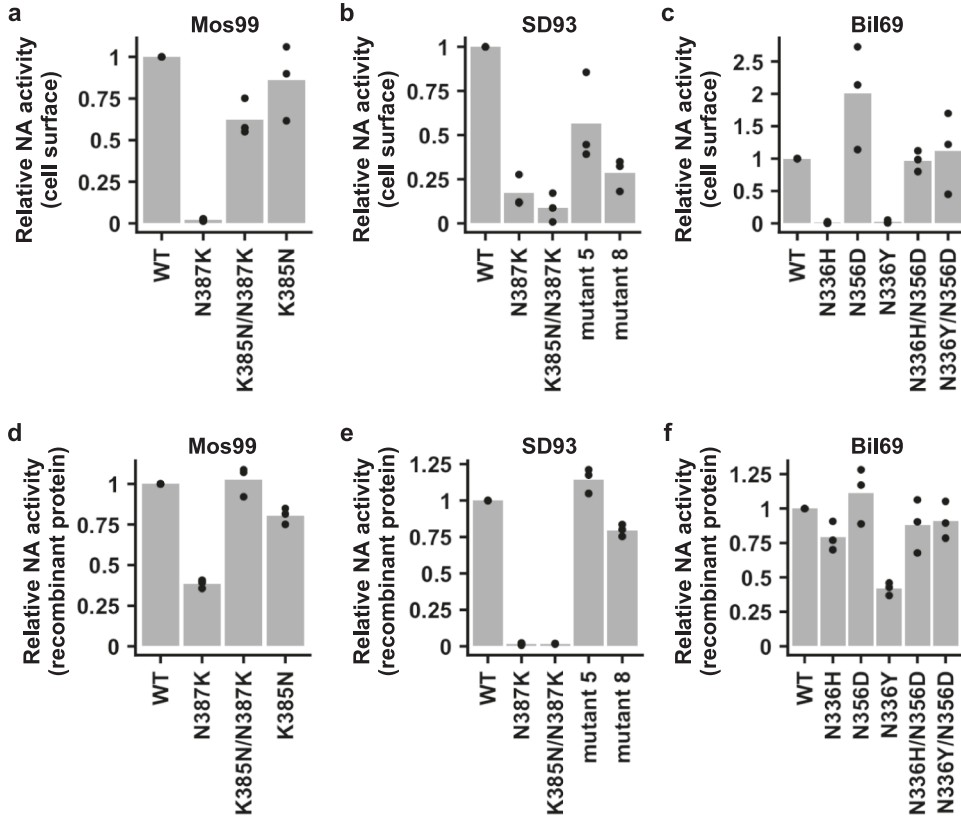

**Fig. 4 | Functional characterization of different NA mutants. a–c** Relative cell surface NA activity of **a** Mos99 mutants, **b** SD93 mutants, and **c** Bil69 mutants. **d–f** Relative NA activity of recombinant proteins of **d** Mos99 mutants, **e** SD93 mutants, and **f** Bil69 mutants. Each bar represents the mean of three independent biological replicates. Each datapoint represents one biological replicate. Relative NA activity of the mutants was normalized to that of their corresponding WT. Source data are provided as a Source data file.

of different mutants was largely consistent with their effects on virus replication fitness (Figs. 2b, e and 3c).

We further measured the NA activity of different mutants using recombinant proteins (Fig. 4d–f). While the NA activity of recombinant proteins showed an overall similar pattern as cell surface NA activity, there were two notable differences. First, although N387K almost completely abolished the cell surface NA activity of Mos99, it still retained around 40% NA activity as a recombinant protein. Similarly, although N336H and N336Y completely abolished the cell surface NA activity of Bil69, they still retained around 75% and 40% NA activity, respectively, as recombinant proteins. These results indicate that protein enzymatic activity can only partly explain the epistasis among natural mutations in human H3N2 NA.

### N387K lowers the membrane trafficking efficiency of NA
Since a high cell surface NA activity would require efficient membrane trafficking of NA, we hypothesized that membrane trafficking also played a role in the epistasis among NA mutations. Flow cytometry analysis showed that N387K decreased the cell surface expression of Mos99 NA, while the addition of K385N restored it (Fig. 5a and Supplementary Fig. 4a). Microscopy analysis further illustrated that the correlation between the signals of Calnexin, which is an endoplasmic reticulum (ER) maker, and NA was higher in N387K ($r = 0.84$) than WT ($r = 0.43$) and K385N/N387K ($r = 0.67$), indicating that N387K increased the endoplasmic reticulum (ER) retention of Mos99 NA (Fig. 5b). In contrast, none of Mos99 NA WT, N387K, and K385N/N387K displayed strong Golgi localization (Supplementary Figs. 5 and 4b). These results demonstrate that membrane trafficking efficiency also contributes to the epistasis in the evolution of human H3N2 NA.

### Specific epistasis due to non-additive effects on protein stability
To dissect the biophysical basis of epistasis, we quantified the protein stability of different NA mutants by measuring their melting temperatures ($T_m$) using thermal shift assay. Our results showed that N387K reduced the $T_m$ of Mos99 NA and SD93 NA by around 5 °C (Fig. 6a). The destabilizing effect of N387K could be partially compensated by K385N. However, K385N alone was not stabilizing, suggesting that the epistasis between K385N and N387K was specific. These results suggest that the defect of N387K in trafficking and enzymatic activity can be attributed to the decrease in protein stability.

In contrast to N387K, N336H did not reduce the $T_m$ of Bil69 NA. However, the first differential curve of the thermal shift assay for Bil69 N336H had an atypical shape (Supplementary Fig. 6). Similarly, the first differential curve for Bil69 N336Y showed two peaks. These results indicate the presence of multiple oligomeric states, hence protein aggregates[20]. The shapes of their first differential curves returned to WT-like when N356D was introduced. As a result, the biophysical basis of epistasis between K385N and N387K is different from that between N336H/Y and N356D.

To further examine the structural mechanism of epistasis between K385N and N387K, we determined the crystal structure of NAs from Bil69, SD93, and Mos99 to 1.54 Å, 1.65 Å, and 1.40 Å, respectively (Supplementary Table 1). Of note, the structure of SD93 NA was determined in complex with zanamivir. Asn387 in Mos99 NA is sandwiched by two positively charged residues, Lys385 and Lys389 (Fig. 6b). Consequently, the positively charged mutation N387K would result in an unfavorable electrostatic interaction with Lys385 and Lys389, which explains the destabilizing effect of N387K in Mos99 NA. The backbone conformation of the loop that contains residues 385 to 389 is highly conserved between NAs from Bil69, SD93, and Mos99

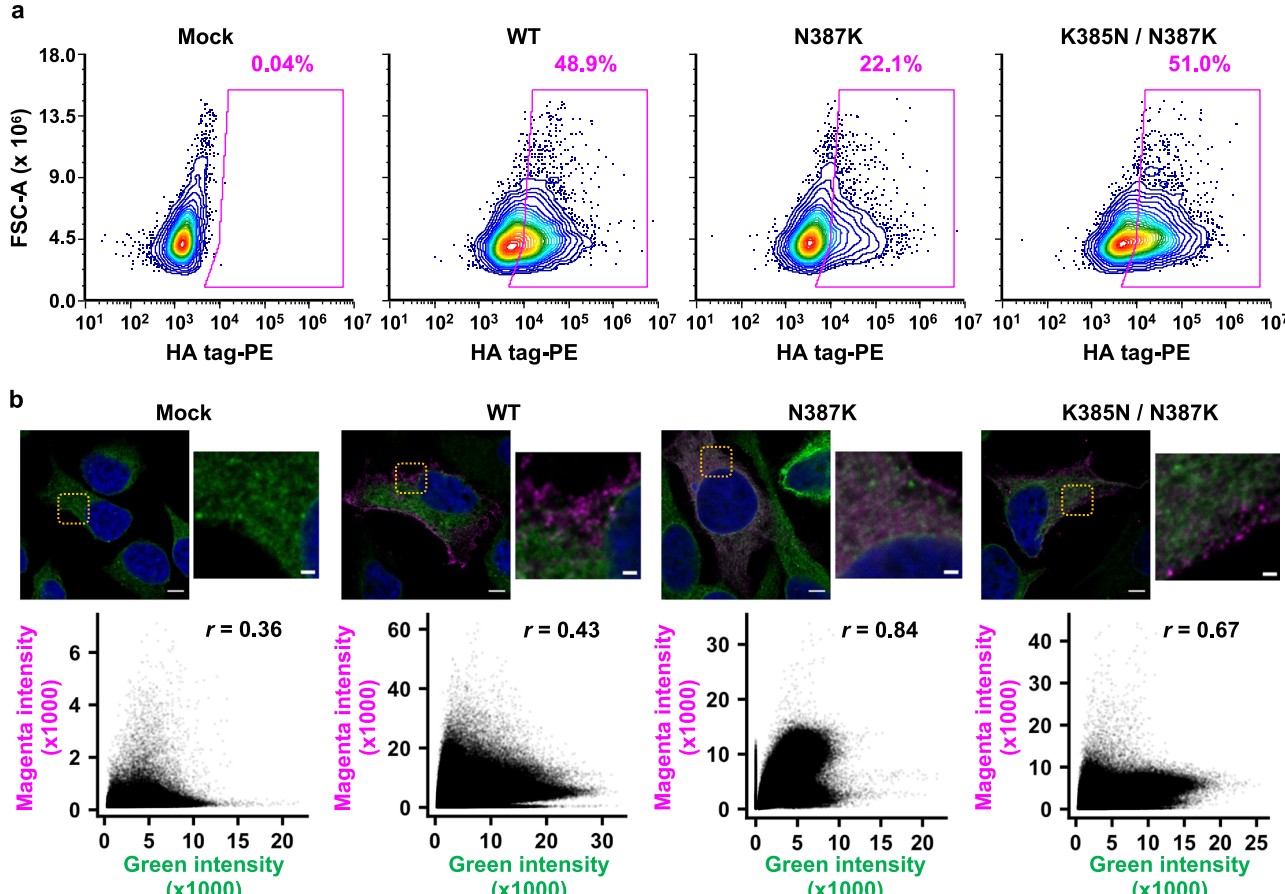

**Fig. 5 | Cell surface protein expression and intracellular localization of NA Mos99 mutants. a** Cell surface expression of HA-tagged Mos99 mutants was analyzed by flow cytometry. **b** Confocal microscopy result of HA-tagged Mos99 mutants. Blue (DAPI), Green (Calnexin, ER), Magenta (NA). The orange box highlights the zoomed in region. Scale bar for the large image is 5 μm and for zoomed in image is 2 μm. Below each micrograph is a cytofluorogram, in which each datapoint represents a pixel. The Pearson correlation coefficient (r) between the green intensity (Calenxin, ER) and magenta intensity (NA) across all pixels in each image is indicated. The result of a representative experiment out of two independent replicates is shown.

(Fig. 6c)[21]. However, the backbone of this loop has a ~3 Å shift in the NAs of more recent human H3N2 strains, namely A/Perth/16/2009 (Perth09) and A/Tanzania/205/2010 (Tan10) (Fig. 6d, e)[22,23]. This shift can be explained by the double mutant N387K/K385N, which is naturally present in Perth09 NA and Tan10 NA. K385N alone is unlikely to cause a change in backbone conformation, since Bil69 NA and Mos99 NA carry different amino acids at residue 385 and yet have the same backbone conformation in this region. Nevertheless, K385N removes a positively charged side chain, which enables Lys387 to adapt a different backbone conformation and point away from Lys389 to minimize unfavorable electrostatic interaction. Therefore, our structural analysis explains why K385N is a permissive mutation for N387K and exemplifies how specific epistasis in the natural evolution of human H3N2 NA modulates its structural conformation.

**Natural relevance of the epistatic network**
At the end, we analyzed the natural evolution of residues in the epistatic network (Fig. 6f). One immediate observation was that N336Y and N387K arose shortly after the emergence of their corresponding permissive mutations. Specifically, N336Y emerged and reached a high occurrence frequency (31%) in 1971, which was only one year after its permissive mutation N356D reached fixation in 1970. Similarly, the occurrence frequency of N387K increased from 3% to 76% during the 2006–2007 influenza season, shortly after its permissive mutation K385N reached fixation in 2005. We also noticed that mutation K369T emerged in 2009–2010 influenza season, which was within 2 years after the fixation of N387K. N387K added a positively charged residue

on the NA surface, whereas K369T did the opposite. As a result, it is possible that N387K promoted the emergence of K369T due to charge balancing[9].

Several residues in the epistatic network are within or immediately adjacent to known antigenic regions. For example, residues 336 and 369 are at antigenic regions I and III[24,25], respectively, whereas residues 248 and 249 are immediately adjacent to a recently identified epitope[26]. Moreover, mutations at residue 344, which locates at antigenic region II, can lead to antibody escape[24,27]. These observations substantiate the involvement of epistasis in NA antigenic drift, which corroborates with our recent study on a seven-residue antigenic region that contains residues 344 and 369[9].

## Discussion
Epistatic interaction between mutations is a key determinant of adaptability and evolutionary trajectory[13,15,28]. Through a systematic analysis, this study described the prevalence of epistasis in the evolution of human H3N2 NA and identified an extensive epistatic network that contains residues 248, 249, 336, 344, 356, 369, 385, 387. In-depth characterization of the epistatic network further revealed the underlying mechanisms. The complexity of epistasis in human H3N2 NA evolution is demonstrated by the number of mutations in the epistatic network, their widespread locations on the protein, as well as the diverse biophysical mechanisms.

Our recent study shows that charge balancing gives rise to epistasis in the antigenic evolution of NA[9]. Such result is highly parallel to the specific epistasis between K385N and N387K in the present study,

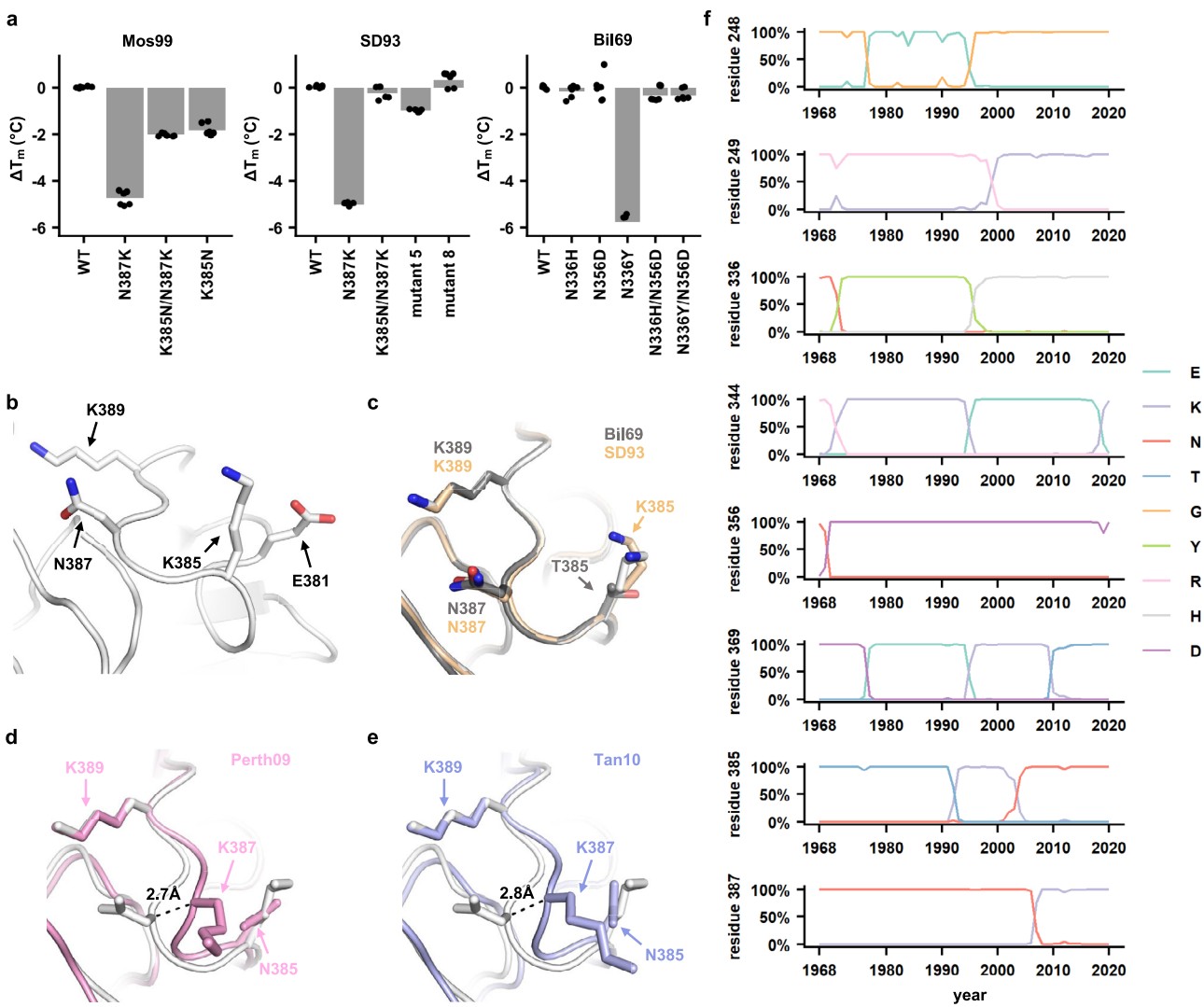

**Fig. 6 | Structure and stability of H3N2 neuraminidase. a** The effects of different Mos99, SD93, and Bil69 NA mutants on protein melting temperature ($T_m$) were measured by thermal shift assay. Each bar represents the mean of six independent biological replicates. Each datapoint represents one biological replicate. Jittering was added to avoid overlap of datapoints. **b** The locations of the residues of interest on Mos99 NA are shown. **c–e** The local structure around residue 387 of Mos99 NA (white) is compared with that of **c** Bil69 NA (gray) and SD93 NA (wheat), **d** H3N2 A/Perth/16/2009 (Perth09) NA (pink, PDB 6BR5)[22], and **e** H3N2 A/Tanzania/205/2010 (Tan10) NA (blue, PDB 4GZO)[23]. **f** Natural occurrence frequencies of the amino acid variants that have a natural occurrence at >50% in any given year at the residues of interest are shown. Source data are provided as a Source data file.

which is governed by non-additive stability effect due to electrostatic interaction. Nevertheless, the epistasis between N336H/Y and N356D has a clearly different mechanism from that between K385N and N387K. Furthermore, wild-type strains with very different NA stability can have similar virus replication fitness (Figs. 1c and 6a), suggesting that protein stability is only one of the biophysical determinants of NA for virus replication fitness. As a result, the biophysical constraints of NA evolution and antigenic drift remain to be fully comprehended.

An interesting observation in this study is that while SD93 NA K385N/N387K mutant has a WT-like thermostability, which indicates proper folding of the protomer, its enzymatic activity is minimal. Since tetramerization of NA is required for enzymatic activity[8,29], it is possible that the head domain of the SD93 NA K385N/N387K mutant failed to form a proper tetramer, despite all recombinant NA proteins in this study being fused to a tetramerization domain and purified as a tetramer by size exclusion chromatography. Consistently, our preliminary analysis on SD93 NA using tryptophan fluorescence spectroscopy indicates that the conformations of N387K and K385N/N387K are different from WT

and the two mutants with WT-like enzymatic activity (Supplementary Fig. 7). Because the surface loop that contains residues 385 and 387 is distal from the tetramerization interface, future studies are needed to explore whether mutations on the NA surface can modulate the tetramerization process through long-range interaction.

Understanding the biophysical constraints of NA evolution not only provides mechanistic insights into influenza antigenic drift, but also facilitates next-generation immunogen design. Development of NA-based immunogens has largely focused on the tetramerization domain[30–32]. In contrast, immunogen designs for other viruses, such as HIV and SARS-CoV-2, mainly focus on introducing mutations to the antigen to increase its stability and expression yield[33–37]. The feasibility of incorporating stabilizing mutations into NA-based immunogen design is demonstrated by a mutation in the center of an N9 NA that improves tetramerization[38]. As NA is an emerging target for the development of a universal influenza vaccine[7,39,40], knowledge of the sequence-structure-function relationship for NA is becoming more important than ever.

## Methods

### Virus rescue experiments

Except A/Udorn/1972 (Udorn72), all H3N2 viruses in this study were generated using the influenza A/WSN/33 (H1N1) eight-plasmid reverse genetics system[41]. Chimeric 6:2 reassortants were employed with six internal segments (PB2, PB1, PA, NP, M, and NS) from A/WSN/33. For HA, the ectodomain was from HK68, whereas the non-coding region, N-terminal secretion signal, C-terminal transmembrane domain, and cytoplasmic tail were from A/WSN/33[16]. For NA, the entire coding region was from the H3N2 strain of interest, whereas the non-coding region of NA was from A/WSN/33[16]. Transfection was performed in a co-culture of HEK 293T and MDCK-SIAT1 cells (ratio of 6:1) at 60% confluence using Lipofectamine 2000 (Thermo Fisher Scientific) according to the manufacturer's instructions. At 24 h post-transfection, cells were washed with phosphate-buffered saline (PBS) and cell culture medium was replaced with OPTI-MEM medium supplemented with 0.8 μg mL$^{-1}$ TPCK-trypsin. Virus was harvested at 72 h post-transfection. To measure virus titer by TCID$_{50}$ assay, MDCK-SIAT1 cells were washed with PBS before the addition of virus in OPTI-MEM medium supplemented with 0.8 μg mL$^{-1}$ TPCK-trypsin.

For the rescue of the authentic Udorn72, a 12-plasmid reverse genetics system was used. The Udorn72 reverse genetics plasmids were gifts from Kanta Subbarao (Doherty Institute). The Udorn72 HA segment-encoding plasmids used were found to have the following mutations: A81G (N18D), C129T (H34Y), G1103T (silent), T1486A (F486Y), and A1614G (N529D) relative to the Udorn72 HA reference sequence (GenBank: AX350190). The Udorn72 NA segment-encoding plasmids used were found to have the following mutations: G992A (A325P) and C1216G (D399E) relative to the Udorn72 NA reference sequence (GenBank: CY009638). Viruses were rescued by transfecting HEK 293T cells using JetPrime (Polyplus) according to the manufacturer's instructions. At 24 h post-transfection, the medium was replaced with viral growth medium (MEM, 1 mM HEPES, 1 μg mL$^{-1}$ of TPCK-trypsin, and 50 μg mL$^{-1}$ of gentamicin) containing $2 \times 10^5$ MDCK cells. Transfection supernatants were collected at 24 h after medium change. Transfection supernatants were titered by plaque assay in MDCK cells.

### Construction of the SD93 NA combinatorial mutant library

The SD93 NA-encoding plasmid from the influenza eight plasmid reverse genetics system was used as the template for insert and vector PCRs. Primers are shown in Supplementary Table 2. To generate the insert, two independent PCRs were performed. The first PCR used SD93lib-I265T-F and SD93lib-336-346-5mut-R as primers. The second PCR used SD93lib-336-346-5mut-F and SD93lib-E369K-R as primers. The products of these two PCRs were mixed and used as templates for an overlapping PCR with SD93lib-E248G-R249K-F and SD93lib-G381E-R as primers. The product of the overlapping PCR was the complete insert. To generate the vector, SD93lib-N387K-VF and SD93lib-VR were used as primers. All PCRs were performed using PrimeSTAR Max polymerase (Takara Bio) according to the manufacturer's instructions. PCR products were purified using Monarch DNA Gel Extraction Kit (New England Biolabs). Both the vector and the complete insert were digested by BsmBI (New England Biolabs) and ligated using T4 DNA ligase (New England Biolabs). The ligated product was transformed into MegaX DH10B T1R cells (Thermo Fisher Scientific). At least half million colonies were collected. The plasmid mutant library was purified from the bacteria colonies using PureLink HiPure Expi Plasmid Purification Kit (Thermo Fisher Scientific).

### Construction of the Bil69 NA combinatorial mutant library

The Bil69 NA-encoding plasmid from the influenza eight plasmid reverse genetics system was used as the template for insert and vector PCRs. Primers are shown in Supplementary Table 3. To generate the insert, three independent PCRs were performed. The first PCR (PCR #1) used Bil69lib-R249K-F and Bil69lib-D286G-R as primers. The second PCR (PCR #2) used Bil69lib-I302V-M307V-F as the forward primer and a mixture of Bil69lib-D329N-K334S-1-R and Bil69lib-D329N-K334S-2-R at equal molar ratio as the reverse primer. The third PCR (PCR #3) used a mixture of Bil69lib-D329N-K334S-1-F and Bil69lib-D329N-K334S-2-F at equal molar ratio as the forward primer, and Bil69lib-N356D-R as the reverse primer. The product of PCR #1 was used as a template for another PCR (PCR #4) using Bil69lib-R249K-F and Bil69lib-I302V-M307V-R as primers. The products of PCRs #2, #3, and #4 were mixed and used as templates for an overlapping PCR with Bil69lib-R249K-F and Bil69lib-L370S-R as primers. The product of the overlapping PCR was the complete insert. To generate the vector, Bil69lib-VF and Bil69lib-VR were used as primers. All PCRs were performed using PrimeSTAR Max polymerase (Takara Bio) according to the manufacturer's instructions. PCR products were purified using Monarch DNA Gel Extraction Kit (New England Biolabs). Both the vector and the complete insert were digested by BsmBI (New England Biolabs) and ligated using T4 DNA ligase (New England Biolabs). The ligated product was transformed into MegaX DH10B T1R cells (Thermo Fisher Scientific). At least half million colonies were collected. The plasmid mutant library was purified from the bacteria colonies using PureLink HiPure Expi Plasmid Purification Kit (Thermo Fisher Scientific).

### Characterizing the fitness of individual variants in the mutant libraries

Virus mutant libraries were rescued in a co-culture of HEK 293T and MDCK-SIAT1 cells (ratio of 6:1) at 60% confluence in a T75 flask (75 cm$^2$) using Lipofectamine 2000 (Thermo Fisher Scientific) according to the manufacturer's instructions. At 24 h post-transfection, cells were washed twice with PBS and cell culture medium was replaced with OPTI-MEM medium supplemented with 0.8 μg mL$^{-1}$ TPCK-trypsin. Virus mutant libraries were harvested at 72 h post-transfection. Virus mutant libraries were titered by TCID$_{50}$ assay using MDCK-SIAT1 cells and stored at −80 °C until used. To passage the virus mutant libraries, MDCK-SIAT1 cells in a T75 flask were washed twice with PBS and infected with half of the virus stock in OPTI-MEM medium supplemented with 0.8 μg mL$^{-1}$ TPCK-trypsin. At 2 h post-infection, infected cells were washed twice with PBS and fresh OPTI-MEM medium supplemented with 0.8 μg mL$^{-1}$ TPCK-trypsin was added to the cells. At 24 h post-infection, supernatant was harvested. Each replicate was transfected and passaged independently. Viral RNA was then extracted from the supernatant using QIAamp Viral RNA Mini Kit (Qiagen). The extracted RNA was reverse transcribed to cDNA using Superscript III reverse transcriptase (Thermo Fisher Scientific). The plasmid mutant libraries and the cDNA from the post-infection viral mutant libraries were amplified by PCR to add part of the adapter sequence required for Illumina sequencing. For SD93 mutant library, SD93lib-recover-F and SD93lib-recover-R were used as primers (Supplementary Table 2). For Bil69 mutant library, Bil69lib-recover-F and Bil69lib-recover-R were used as primers (Supplementary Table 3). A second PCR was carried out to add the rest of the adapter sequence and index to the amplicon using primers: 5′-AAT GAT ACG GCG ACC ACC GAG ATC TAC ACT CTT TCC CTA CAC GAC GCT-3′ and 5′-CAA GCA GAA GAC GGC ATA CGA GAT XXX XXX GTG ACT GGA GTT CAG ACG TGT GCT-3′. Positions annotated by an "X" represented the nucleotides for the index sequence. The final PCR products were submitted for the next-generation sequencing using Illumina MiSeq PE250.

### Sequencing data analysis

Sequencing data were obtained in FASTQ format. Forward and reverse reads were merged by PEAR[42]. Using the SeqIO module in BioPython[43], primer sequences were trimmed from the merged reads. If the length of a given trimmed read did not match with that of the reference nucleotide sequence, the read would be discarded. The trimmed reads were then translated to amino acid sequences. Amino acid mutations

were called by comparing the translated reads to the translated reference sequence. For Bil69 mutant library, frequency of mutant $i$ in sample $s$ was computed for each replicate as follows:

$$\text{frequency}_{i,s} = \frac{\text{read count}_{i,s} + 1}{\sum_{k \in s} \text{read count}_{k,s}} \qquad (1)$$

A pseudocount of 1 was added to the read counts to avoid division by zero in subsequent steps. Subsequently, the enrichment of mutant $i$ was computed for each replicate as follows:

$$\text{enrichment}_i = \frac{\text{frequency}_{\text{post-passaged library}}}{\text{frequency}_{\text{plasmid mutant library}}} \qquad (2)$$

Bil69 mutants with a read count of less than 10 in the plasmid mutant library were discarded. For SD93 mutant library, the frequency of each mutant was calculated without the addition of pseudocount.

## Protein expression and purification

The NA head domains, which contained residues 82 to 469, were fused to an N-terminal gp67 signal peptide, $6 \times$ His-tag, a vasodilator-stimulated phosphoprotein (VASP) tetramerization domain, and a thrombin cleavage site. Recombinant bacmid DNA that carried the NA ectodomain from the strain of interest was generated using the Bac-to-Bac system (Thermo Fisher Scientific) according to the manufacturer's instructions. Baculovirus was generated by transfecting the purified bacmid DNA into adherent Sf9 cells using Cellfectin reagent (Thermo Fisher Scientific) according to the manufacturer's instructions. The baculovirus was further amplified by passaging in adherent Sf9 cells at a multiplicity of infection (MOI) of 1. Recombinant NA head domains were expressed by infecting 1 L of suspension Sf9 cells at an MOI of 1. On day 3 post-infection, Sf9 cells were pelleted by centrifugation at $4000 \times g$ for 25 min, and soluble recombinant NA was purified from the supernatant by affinity chromatography using Ni Sepharose excel resin (Cytiva) and then size exclusion chromatography using a HiLoad 16/100 Superdex 200 prep grade column (Cytiva) in 20 mM Tris-HCl pH 8.0, 100 mM NaCl, and 10 mM $CaCl_2$. For crystallography, recombinant NA was further digested by thrombin (MilliporeSigma) for three weeks in 4 °C using 15 U thrombin per mg of recombinant NA. The thrombin-digested recombinant NA was incubated with TALON metal affinity resin (Takara) for 2 h. The thrombin-digested recombinant NA in the flow-through and 10 mM imidazole wash was purified by size exclusion chromatography using a HiLoad 16/100 Superdex 200 prep grade column (Cytiva) in 20 mM Tris pH 8.0, 100 mM NaCl, and 10 mM $CaCl_2$.

## Cell surface NA activity assay

NA from the strain of interest was cloned into the phCMV3 vector. phCMV3-NA was transfected into 293T cells using Lipofectamine 2000 (Thermo Fisher Scientific). At 24 h post-transfection, the cells were washed with PBS and then resuspended in 200 μL PBS. Cells were then mixed with 800 μM fluorogenic substrate methylumbelliferyl-α-D-N-acetylneuraminic acid (MUNANA) (MilliporeSigma) in PBS that contained 33 mM MES pH 6.5 and 4.0 mM $CaCl_2$. The sample was then transferred to a 96-well half area plate (Corning). NA enzymatic kinetics at 37 °C were measured by a SpectraMax M2 microplate reader (Molecular Devices) with an excitation at 365 nm and an emission at 450 nm. Fluorescence was recorded for 50 min at 30 s interval. NA activity was quantified by the rate of increase in fluorescence signal. Background signal, which was determined by untransfected cells, was subtracted. NA activity was normalized to the cell concentration, which was determined by a Countess II FL Automated Cell Counter (Thermo Fisher Scientific).

## Recombinant NA activity assay

In a 96-well half area plate (Corning), 2.5 ng of recombinant NA was incubated with 100 μM MUNANA in PBS that contained 33 mM MES pH 6.5 and 4 mM $CaCl_2$. NA enzymatic kinetics at 37 °C were measured by a SpectraMax M2 microplate reader (Molecular Devices) with an excitation at 365 nm and an emission at 450 nm. Fluorescence was recorded for 30 min at 30 s interval. NA activity was quantified by the rate of increase in fluorescence signal. Background signal, which was determined in the absence of recombinant NA, was subtracted.

## Flow cytometry analysis of NA expression

$4 \times 10^5$ HEK 293T cells were seeded in 6-well plates and grown overnight at 37 °C and 5% $CO_2$ in a humidified incubator. Cells were transfected with 2 μg of phCMV3 plasmids encoding the C-terminal HA-tagged NA from the strain of interest using Lipofectamine 2000 (Thermo Fisher Scientific) according to the manufacturer's instructions. At 24 h post-transfection, medium was discarded, and cells were washed once with PBS. Cells were subsequently detached with versene (Thermo Fisher Scientific) and resuspended in fluorescence-activated single cell sorting (FACS) buffer (50 mM EDTA, 2% v/v FBS in DMEM with high glucose and HEPES, without phenol red). Cells were then pelleted via centrifugation at $300 \times g$ for 5 min at 4 °C and the supernatant was aspirated. In the subsequent steps, cells and all reagents were kept on ice.

For surface staining, cell pellets were resuspended in 1 mL FACS buffer. Cells were incubated with 1 μg mL$^{-1}$ PE anti-HA.11 (BioLegend, catalog #: 901518) at 4 °C with rocking for 1 h. Subsequently, cells were pelleted via centrifugation at $300 \times g$ for 5 min at 4 °C and the supernatant was aspirated. Cells were washed with 1 mL FACS buffer, pelleted, and resuspended in 1 mL FACS buffer before flow cytometry analysis.

For intracellular staining, cell pellets were fixed in 300 μL of 1% v/v paraformaldehyde (in PBS) and incubated at 4 °C with rocking for 30 min. Cells were pelleted, washed with 1 mL FACS buffer, permeabilized in 300 μL of 0.1% v/v Triton X-100 (in PBS) and incubated at 4 °C with rocking for 30 min. Cells were pelleted, washed with 1 mL FACS buffer and incubated with 1 μg mL$^{-1}$ PE anti-HA.11 (BioLegend, catalog #: 901518) at 4 °C with rocking for 1 h. Then, cells were pelleted, washed with 1 mL FACS buffer, and resuspended in 1 mL FACS buffer before flow cytometry analysis.

For flow cytometry analysis, cells were analyzed using an Accuri C6 flow cytometer (BD Biosciences) with a 488 nm laser and a 585/40 bandpass filter. Data were collected using Accuri C6 software. $10^4$ singlets were collected for each sample. Data were analyzed using FCS Express 6 software (De Novo Software).

## Confocal microscopy

$7.5 \times 10^4$ HeLa cells were seeded on 12 mm #1.5 glass cover slips in 24-well plates and grown overnight at 37 °C with 5% $CO_2$ in a humidified incubator. Then, cells were transfected using Lipofectamine 2000 (Thermo Fisher Scientific) according to the manufacturer's instructions. At 24 h post-transfection, medium was discarded and cells were washed with PBS. Cells were fixed with 4% paraformaldehyde for 10 min at room temperature and then washed with PBS. Cells were permeabilized with ice-cold 0.1% v/v Triton X-100 (in PBS) for 20 min at 4 °C and subsequently washed with PBS. Cells were blocked with 5% v/v normal donkey serum (in PBS) for 1 h at room temperature. Then, cells were incubated overnight at 4 °C with the indicated primary antibody, namely mouse anti-HA tag (Thermo Fisher Scientific, catalog #: 26183) 1:500, rabbit anti-calnexin (Thermo Fisher Scientific, catalog #: PA5-34754) 1:500, or rabbit anti-GM130 (Thermo Fisher Scientific, catalog #: 11308-1-AP) 1:200, and then washed with PBS. Subsequently, cells were incubated for 1 h at room temperature with secondary antibody, namely Alexa Fluor 488 donkey anti-rabbit IgG (Abcam, catalog #: ab150073) 1:200 or Alexa Fluor 555 donkey anti-mouse IgG (Thermo

Fisher Scientific, catalog #: A-31570) 1:500, as well as DAPI (to a final concentration of $1\,\mu g\,ml^{-1}$), and then washed with PBS. All primary antibodies, secondary antibodies, and DAPI were diluted in 5% v/v normal donkey serum (Abcam). Cover slips were mounted on slides with ProLong Diamond Antifade Mountant (Thermo Fisher Scientific) and sealed with nail polish.

Images were captured on an LSM 700 microscope and a Plan Apochromat 63X/1.4 Oil DIC objective using Zen software (Zeiss). Excitation laser lines 405 nm, 488 nm, and 555 nm were used for acquiring images stained with DAPI, Alexa Fluor 488, and Alexa Fluor 555, respectively. Individual channels of each image were linearly adjusted using Zen software. Transfected cells were manually outlined as regions of interest in Fiji. Cytofluorogram data and Pearson correlation coefficients of transfected cells were calculated with fluorescence intensity thresholds automatically adjusted using the JACoP plug-in[44].

### Thermal shift assay
5 µg protein was mixed with 5× SYPRO orange (Thermo Fisher Scientific) in 20 mM Tris-HCl pH 8.0, 100 mM NaCl, and 10 mM $CaCl_2$ at a final volume of 25 µL. The sample mixture was then transferred into an optically-clear PCR tube (VWR). SYPRO orange fluorescence data in relative fluorescence unit (RFU) was collected from 10 °C to 95 °C using a CFX Connect Real-Time PCR Detection System (Bio-Rad). The temperature corresponding to the lowest point of the first derivative $-d(RFU)/dT$ was determined to be the $T_m$.

### Crystallization and structural determination
Crystallization screening was performed using the JCSG Core Suites I-IV (Rigaku) with thrombin-digested NA at 7 mg mL$^{-1}$. Sitting drop for crystallization screening was set up by equal volume of precipitant and protein solution using the Crystal Gryphon (Art Robbins Instruments). Crystallization screens were incubated at 18 °C. Initial hits were further optimized using the sitting drop method at 18 °C, with 350 µL reservoir solution and 1:1 ratio of precipitant and protein solution. The crystallization conditions were as follows:

Mos99 WT: 2.0 M ammonium sulfate, 0.1 M acetate pH 4.6
SD93 WT + zanamivir: 8% PEG-8000, 0.1 M Tris pH 8.5
Bil69 WT: 10% PEG-6000, 5% MPD, 0.1 M HEPES pH 6.5

Crystals were soaked in precipitant solution containing the cryoprotectant prior to vitrification in liquid nitrogen. The cryoprotectants were as follows:

Mos99 WT: 15% ethylene glycol
SD93 WT + zanamivir: 25% PEG 200
Bil69 WT: 20% ethylene glycol

Data were collected at the Advanced Photon Source (APS) at Argonne National Laboratory via the Life Science Collaborative Access Team (LS-CAT) at beamlines 21-ID-D, 21-ID-G, and 21-ID-F. Initial diffraction data were indexed, integrated, and scaled using autoPROC[45]. The structures were solved by molecular replacement using Phaser-MR included in the Phenix suite[46], using PDB 2AEP as the replacement model[47]. The structures were further refined using REFMAC5[48] and were manually built in COOT[49]. Ramachandran statistics were calculated using MolProbity[50].

### Fluorescent emission spectroscopy
The steady-state tryptophan emission spectra were measured using a fluorometer (FP8300, JASCO). Samples for the emission spectra measurements were prepared by diluting the protein stock in SEC buffer (20 mM Tris-HCl pH 8.0, 100 mM NaCl, 10 mM $CaCl_2$) to an $OD_{280} = 0.32$ as measured by UV-Vis spectroscopy (UV-1800, Shimadzu). The Trp residues in the protein samples were excited at 295 nm and the emission spectra were collected from 290 to 450 nm with 2.5 nm bandwidths. The emission spectra data were then normalized to a range of 0 to 1 and plotted for analysis.

### Statistics and reproducibility
No statistical method was used to predetermine sample size. All observations reported in this manuscript were made in at least two independent experiments, typically with biological triplicates. No data were excluded from the analyses. The Investigators were not blinded to allocation during experiments and outcome assessment.

### Reporting summary
Further information on research design is available in the Nature Research Reporting Summary linked to this article.

## Data availability
Raw sequencing data generated in this study have been deposited in the NIH Short Read Archive under accession number: BioProject PRJNA790468. The X-ray coordinates and structure factors have been deposited in the RCSB Protein Data Bank under accession codes 7U4E, 7U4F, and 7U4G. Source data are provided with this paper.

## Code availability
Custom python scripts for analyzing the next-generation sequencing data have been deposited to https://github.com/nicwulab/N2_evol_contingency[51].

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

## Acknowledgements

This work was supported by National Institutes of Health (NIH) R00 AI139445 (N.C.W.), DP2 AT011966 (N.C.W.), and R01 AI167910 (N.C.W.). We thank Martin Gruebele for helpful discussion, the Roy J. Carver Biotechnology Center at the University of Illinois at Urbana-Champaign for assistance with next-generation sequencing, the Microscopy Core Facility of the Institute of Genomic Biology at the University of Illinois at Urbana-Champaign for microscopy use, and Spencer Anderson and colleagues at LS-CAT (Argonne National Labs) for facilitating X-ray data collection.

## Author contributions

R.L. and N.C.W. conceived and designed the study. R.L., M.D., Z.T.D., and C.B.B. performed the virus rescue experiments. R.L. constructed the mutant libraries and prepared the sequencing libraries. Y.W. and N.C.W. analyzed the next-generation sequencing data. T.J.C.T. and Q.W.T. performed the flow cytometry and microscopy experiments. R.L., C.S.G., and C.T. expressed and purified the recombinant NA. R.L. and

C.S.G. performed the MUNANA assay and the thermal shift assay. R.L., A.H.G., and S.K.N. performed protein crystallization. A.H.G. and S.K.N. collected the X-ray diffraction data as well as determined and refined the structures. R.L. and G.G. performed the tryptophan fluorescence spectroscopy. R.L. and N.C.W. wrote the paper and all authors reviewed and edited the paper.

## Competing interests

N.C.W. serves as a consultant for HeliXon. The authors declare no other competing interests.

## Additional information

**Correspondence and requests** for materials should be addressed to Nicholas C. Wu.

