## [Peer Review File · Nature Communications]

Prevalence and mechanisms of evolutionary contingency in human influenza H3N2 neuraminidaseReviewers' comments:

Reviewer #1 (Remarks to the Author):

In this work, Ruipeng Lei and colleagues investigate the mutational fitness effects and potential epistatic interactions within the H3N2 NA protein. They identify two mutations (N387K and N336H) which are deleterious in an ancestral strain, but nonetheless persist in more recent strains of H3N2. Through analysis of NA enzymatic function, protein stability, and protein structure, they find that N387K decreases the stability of NA and thus requires a compensatory stabilizing mutation (K385N) to restore viral fitness. They also find that N336H may impact oligomerization of NA, thus requiring the compensatory mutation N356D. This study provides important insight into the mutational fitness effects within NA of H3N2, while paying particular interest to epistatic relationships. The combination of cell culture assays for viral fitness and NA function, with in-depth protein structure and stability analysis reveals an interesting mechanism of charge-balancing to explain the epistatic relationship between N387K and K385N. I have no major concerns regarding this work.

Minor Comments:

1. All of the experiments are performed with 6+2 viruses containing HA/NA of these H3N2 strains along with the remaining segments from WSN33. This is generally fine, as it allows the authors to focus on NA and perhaps its epistatic partner, HA. However, it is possible that there are epistatic modifiers elsewhere in the genome. It would be helpful to test just a few of the key mutants in an authentic H3N2 backbone. This seems feasible given that the HA and NA constructs of interest are in hand, and one would need to do the rescue assay with the other 6 segments from an H3N2, likely an ancestral 1968 strain.
2. The introduction of Y336H on line 145 could be a little more clear and highlight the connection to N336H. The casual reader might not catch that this is a mutation at the same site of one of the deleterious mutations highlighted at the outset (I missed it the first time!). This is important, both in terms of the flow of the paper, and also in terms of the significance of this site in epistatic interactions (e.g. since it arose in the mutagenesis experiment).
3. The authors use different metrics and scales in Figure 2D and 3B, which show the data on the minimal mutational scanning libraries. This needs some clarification and discussion. I assume that this reflects the effect size of mutations, but as presented, the data don't really make sense with respect to each other (e.g. one is log₁₀ frequency with a couple orders of magnitude change and the other is log₁₀ enrichment with a 2-3 fold change).
4. Interpretation of Figure 5B needs more explanation. What is the conclusion from the cytofluorogram? What is each point? It is not mentioned in the text. Also, the panels on this figure should be larger
5. Some typos/grammar issues throughout, particularly in the methods sections
6. Font in figures needs to be larger

Reviewer #2 (Remarks to the Author):

Lei and colleagues identify epistatic relationships that govern the evolution of specific sites within the influenza neuraminidase (NA) gene. Using natural variation, they identify positions where mutations were tolerated in one genetic context and are deleterious in another. They focus on two sites and characterize their genetic dependencies in genetic, molecular, biophysical, and structural detail. Despite physical distance from the enzymatic active site or interfaces between protomers specific mutations disrupt enzymatic activity and trafficking through the secretory pathway. These observations inform our understanding of NA evolution and may aid in designing immunogens that elicit broadly protective immunity.

As presented it is unclear if these observations are generalizable or apply to a very select few positions. As mutations 336 arose during a screen for permissive mutations for N387K (present in all

“fit” viruses) and is part of a second “epistatic network”, these observations may be limited to a single (or few) “choice” site(s). The manuscript could be written as the evolutionary contingency of position 336. As 16% of the protein has changed since 1968, combinatorial networks of mutations are unlikely to govern most of NA’s evolution. For these reasons and others listed below this work may be more suited for a less general audience.

The authors seek to identify epistemic networks and use mutant libraries to identify permissive mutations. They use combinatorial libraries to identify sites that restore replicative fitness. What is missing throughout this manuscript is a connection to the real-world evolution of NA and its sequential acquisition of mutations at the sites profiled in this manuscript. The “epistatic networks” identified and characterized here may be intrinsic to the biology of NA evolution or an artifact of placing specific mutations into disparate contexts.

A prevailing and reasonable assumption is that NA has evolved to evade protective human immunity. The authors do not place any of their work into this context. Are the profiled positions in key antigenic sites, do these sites evolve rapidly and do any of these mutations influence antibody binding? All of these are fundamental questions that should be addressed.

It would appear that the first deep mutational scan for permissive mutations for N387K was not conducted properly and/or was not designed correctly due to biology uncovered by the authors while performing this work. All viruses with increased replication contained a reversion Y336N, despite limited passage. The sequencing of the mutant library required implementation of a “pseudocount” strategy to avoid dividing by zero. Criteria for enrichment were >0.1% vs. 2-fold for the N336 library. As described, it is unclear whether convenient rescue mutations were identified or if true “epistatic networks” exist.

The characterization of viruses with mutations permissive for the N387K mutation is incomplete. All mutations arose in the presence of Y336N and most mutations were only tested in the presence of N387K. While reversion of this site runs contrary to the author’s hypothesis, the biology of the virus suggests the presence of additional mechanisms or dependency networks.

The examination of the structures is cursory and limited to the more complicated “evolutionary network” related to N387K. If I interpreted the sequence alignment and presented structural panels correctly it would appear that additional substitutions in the strand harboring N387 should contribute to its mobility. This is not discussed. The contributions of the entirety of “mutant 8” mutations that recovered most of its infectivity is not discussed. A structural explanation for evolutionary contingency for N336Y is not discussed.

Minor:

Throughout the manuscript graphs are presented with variable axes. Comparisons between panels are difficult. For instance: results of the mutational scanning are plotted quite differently, infectivity values differ by multiple orders of magnitude across panels of the same figure (e.g. Figures 1 or 4) and the melting temperatures of various HAs differ by as much as 10 degrees in side-by side comparisons.

Reviewers' comments:

Reviewer #1 (Remarks to the Author):

In this work, Ruipeng Lei and colleagues investigate the mutational fitness effects and potential epistatic interactions within the H3N2 NA protein. They identify two mutations (N387K and N336H) which are deleterious in an ancestral strain, but nonetheless persist in more recent strains of H3N2. Through analysis of NA enzymatic function, protein stability, and protein structure, they find that N387K decreases the stability of NA and thus requires a compensatory stabilizing mutation (K385N) to restore viral fitness. They also find that N336H may impact oligomerization of NA, thus requiring the compensatory mutation N356D. This study provides important insight into the mutational fitness effects within NA of H3N2, while paying particular interest to epistatic relationships. The combination of cell culture assays for viral fitness and NA function, with in-depth protein structure and stability analysis reveals an interesting mechanism of charge-balancing to explain the epistatic relationship between N387K and K385N. I have no major concerns regarding this work.

Response: Thank you for the positive comments.

Minor Comments:

1. All of the experiments are performed with 6+2 viruses containing HA/NA of these H3N2 strains along with the remaining segments from WSN33. This is generally fine, as it allows the authors to focus on NA and perhaps its epistatic partner, HA. However, it is possible that there are epistatic modifiers elsewhere in the genome. It would be helpful to test just a few of the key mutants in an authentic H3N2 backbone. This seems feasible given that the HA and NA constructs of interest are in hand, and one would need to do the rescue assay with the other 6 segments from an H3N2, likely an ancestral 1968 strain.

Response: We agree with the reviewer that the suggested experiment is an appropriate control. We have performed this experiment and included the results as Figure S2B in the revised manuscript. The result is also described in the results section of the revised manuscript (lines 105-107):

“Consistently, mutating NA residue 336 to His was neutral in the authentic H3N2 A/Udorn/1972, whereas N387K did not yield any detectable titer (Figure S2B).”

Additional details are provided in the Figure legend:

“The replication fitness of NA N387K and NA Y336H in authentic H3N2 A/Udorn/1972 (all eight segments were from A/Udorn/72) was examined by a virus rescue experiment. Virus titer was measured by plaque forming units (PFU). Error bars indicate the standard deviation of three independent experiments.”

2. The introduction of Y336H on line 145 could be a little more clear and highlight the connection to N336H. The casual reader might not catch that this is a mutation at the same site of one of the deleterious mutations highlighted at the outset (I missed it the first time!). This is important, both in terms of the flow of the paper, and also in terms of the significance of this site in epistatic interactions (e.g. since it arose in the mutagenesis experiment).

Response: Thank you for the suggestion. The introduction of Y336H is elaborated in the revised manuscript (lines 153-158):

“As mentioned above, Y336N represented a reversion in the natural evolution of human H3N2 NA (e.g. reverting from Bil71 to Bil69, **Figure S1D**). In contrast, Y336H was the forward mutation in the natural evolution of human H3N2 NA (e.g. mutating from SD93 to Mos99, **Figure S1C**). Since both Y336N and Y336H enabled SD93 to mutate away from Tyr at residue 336, we also tested whether Y336H could act as a permissive mutation for N387K.”

3. The authors use different metrics and scales in Figure 2D and 3B, which show the data on the mini-mutational scanning libraries. This needs some clarification and discussion. I assume that this reflects the effect size of mutations, but as presented, the data don't really make sense with respect to each other (e.g. one is log₁₀ frequency with a couple orders of magnitude change and the other is log₁₀ enrichment with a 2-3 fold change).

Response: “Enrichment” is typically used as a proxy for mutational fitness in a mutational scanning experiment. To compute the “enrichment” of a given mutant, its frequency information in the input library is needed. However, because a de novo mutation is not present in the input library, “enrichment” cannot be used to quantify the fitness of mutants that carries a de novo mutation. This is clarified in the revised manuscript (lines 139-145):

“Of note, Y336N was not included in the construction of the mutant library, and hence was a de novo mutation that emerged during viral passaging. Interestingly, Y336N represented a reversion to the ancestral stains, including HK68 and Bil69 (**Figure S1A and S1D**). While the fitness of each mutant in a mutant library could typically be estimated from its frequency change between the input mutant library and the post-passaged mutant library (i.e. frequency enrichment)¹⁶, the absence of Y336N in the input mutant library prevented the use of frequency enrichment for fitness measurement.”

4. Interpretation of Figure 5B needs more explanation. What is the conclusion from the cytofluorogram? What is each point? It is not mentioned in the text. Also, the panels on this figure should be larger

Response: The figure legend is improved to clarify the meaning of each data point in Figure 5B:

“Below each micrograph is a cytofluorogram, in which each data point represents a pixel. The Pearson correlation coefficient between the green intensity (Calnexin, ER) and magenta intensity (NA) across all pixels in each image is indicated.”

We have also elaborated our description of the cytofluorogram in the results section of the revised manuscript to clarify the conclusion (lines 214-217):

“Microscopy analysis further illustrated that the correlation between the signals of Calnexin, which is an endoplasmic reticulum (ER) maker, and NA was higher in N387K ($r = 0.84$) than WT ($r = 0.43$) and K385N/N387K ($r = 0.67$), indicating that N387K increased the endoplasmic reticulum (ER) retention of Mos99 NA (Figure 5B).”

All panels in figure 5 are enlarged in the revised manuscript.

5. Some typos/grammar issues throughout, particularly in the methods sections

Response: Proofreading was performed. Typos and grammatical errors were fixed in the revised manuscript.

6. Font in figures needs to be larger

Response: Font in figures is enlarged in the revised manuscript.

Reviewer #2 (Remarks to the Author):

Lei and colleagues identify epistatic relationships that govern the evolution of specific sites within the influenza neuraminidase (NA) gene. Using natural variation, they identify positions where mutations were tolerated in one genetic context and are deleterious in another. They focus on two sites and characterize their genetic dependencies in genetic, molecular, biophysical, and structural detail. Despite physical distance from the enzymatic active site or interfaces between protomers specific mutations disrupt enzymatic activity and trafficking through the secretory pathway. These observations inform our understanding of NA evolution and may aid in designing immunogens that elicit broadly protective immunity.

As presented it is unclear if these observations are generalizable or apply to a very select few positions. As mutations 336 arose during a screen for permissive mutations for N387K (present in all “fit” viruses) and is part of a second “epistatic network”, these observations may be limited to a single (or few) “choice” site(s). The manuscript could be written as the evolutionary contingency of position 336. As 16% of the protein has changed since 1968, combinatorial networks of mutations are unlikely to govern most of NA’s evolution. For these reasons and others listed below this work may be more suited for a less general audience.

Response: We respectfully disagree with this reviewer regarding the interpretation of our results and the impact of our work.

For example, our result in Figure 1A demonstrated that >10% of natural mutations in NA are deleterious for an ancestral strain, providing direct evidence for the importance of epistasis in the evolution of NA. Without epistasis, those natural mutations would have never emerged in circulating human influenza H3N2 virus. This observation contradicts with the reviewer’s comment “combinatorial networks of mutations are unlikely to govern most of NA’s evolution”.

In addition, the involvement of epistasis in the antigenic shift of NA, which is a major focus of global public health (PMID: 29081496), reflects the impact of our findings (see lines 250-252 in the first submitted manuscript, which correspond to lines 270-275 in the revised manuscript):

“Several residues in the epistatic network are within or immediately adjacent to known antigenic regions. For example, residue 369 is at antigenic region III^{25,26}, whereas residues 248 and 249 are immediately adjacent to a recently identified epitope²⁷. Moreover, mutations at residue 344, which locates at antigenic region II, can lead to antibody escape^{25,28}. These observations substantiate the involvement of epistasis in NA antigenic drift, which corroborates with our recent study on a seven-residue antigenic region that contains residues 344 and 369⁹.”

More importantly, our work represents the first study to investigate the prevalence of epistasis in the evolution of influenza NA and provides a detailed biophysical characterization of NA. Given that influenza NA is an emerging influenza vaccine target, our work contributes to, as stated by

this reviewer, “[design] immunogens that elicit broadly protective immunity”, which is a primary interest of the National Institute of Allergy and Infectious Diseases (PMID: 29506129).

The authors seek to identify epistemic networks and use mutant libraries to identify permissive mutations. They use combinatorial libraries to identify sites that restore replicative fitness. What is missing throughout this manuscript is a connection to the real-world evolution of NA and its sequential acquisition of mutations at the sites profiled in this manuscript. The “epistatic networks” identified and characterized here may be intrinsic to the biology of NA evolution or an artifact of placing specific mutations into disparate contexts.

Response: We are unclear why the reviewer said that our manuscript is missing a connection to the real-world evolution of NA. Here are several examples of the connection between real-world evolution of NA and mutations/epistasis in this study:

1. Our mutagenesis experiments were rationally designed based on the NA sequences of natural strains. These include all 70 mutations in Figure 1 and all mutations in the epistatic network that we identified (Figures 2 and 3).
2. As shown in Figure 6F, N387K and N336H arose after the emergence of their corresponding permissive mutations that were identified in our study. This observation demonstrates that our result connects to the sequential acquisition of mutations in real world. Further elaboration was provided in the revised manuscript (lines 261-264):

“Interestingly, the natural emergence of N336Y and N387K occurred shortly after the emergence of their corresponding permissive mutations N356D and K385N, respectively (Figure 6F). This observation supports the notion that epistasis is an important determinant of mutation coevolution in human H3N2 NA⁹.”

3. We have demonstrated the biophysical basis of epistasis, which is a solid proof that our findings are intrinsic to the biology of NA evolution.

A prevailing and reasonable assumption is that NA has evolved to evade protective human immunity. The authors do not place any of their work into this context. Are the profiled positions in key antigenic sites, do these sites evolve rapidly and do any of these mutations influence antibody binding? All of these are fundamental questions that should be addressed.

Response: We respectfully disagree with the reviewer’s comment that “The authors do not place any of their work into [the context of NA antigenic drift].” In our first submitted manuscript, we spent half a paragraph in address this issue (see lines 247-253 in the first submitted manuscript, which correspond to lines 270-275 in the revised manuscript):

“Several residues in the epistatic network are within or immediately adjacent to known antigenic regions. For example, residue 369 is at antigenic region III^{25, 26}, whereas residues 248 and 249 are immediately adjacent to a recently identified epitope²⁷. Moreover, mutations at residue 344, which locates at antigenic region II, can lead to antibody escape^{25,28}. These observations substantiate the involvement of epistasis in NA antigenic drift, which corroborates with our recent study on a seven-residue antigenic region that contains residues 344 and 369⁹.”

This section already addressed the reviewer’s questions on “are the profiled positions in key antigenic sites ... do any of these mutations influence antibody binding?”.

It would appear that the first deep mutational scan for permissive mutations for N387K was not conducted properly and/or was not designed correctly due to biology uncovered by the authors while performing this work. All viruses with increased replication contained a reversion Y336N, despite limited passage. The sequencing of the mutant library required implementation of a “pseudocount” strategy to avoid dividing by zero. Criteria for enrichment were >0.1% vs. 2-fold for the N336 library. As described, it is unclear whether convenient rescue mutations were identified or if true “epistatic networks” exist.

Response: We respectfully disagree with the statement “it is unclear whether convenient rescue mutations were identified or if true “epistatic networks” exist.”

As shown in Figures 2E and 3C, all the results in the deep mutational scan were validated. Specifically, Figure 2E shows that even without the reversion Y336N, the deleterious effect of N387K can be epistatically compensated by E248G/R249K/Y336H/K344E/E369K/K385N, which are all natural mutations. Similarly, Figure 3C shows that while natural mutations N336H and N336Y are highly deleterious, their fitness can be epistatically compensated by N356D. In fact, Figure 3C is the simplest and classic example of epistasis, where N356D is neutral alone but highly beneficial in the presence of N336H/Y. Of note, all these findings were informed by the deep mutational scan.

Despite our first deep mutational scan being useful for identifying permissive mutations for N387K, it does have some limitations as suggested by the reviewer. Such limitations are acknowledged in the revised manuscript (lines 164-166):

“As mentioned above, our mutant library only contained 10 out of 18 mutations that differ between SD93 and Mos99. It is possible that some of the remaining eight mutations could also help compensate the fitness defect of N387K.”

The characterization of viruses with mutations permissive for the N387K mutation is incomplete. All mutations arose in the presence of Y336N and most mutations were only tested in the presence of N387K. While reversion of this site runs contrary to the author’s hypothesis, the biology of the virus suggests the presence of additional mechanisms or dependency networks.

Response: For identifying permissive mutations for N387K, we actually did not only test Y336N, which was a reversion, but also the forward mutation Y336H at the same residue. This part of the result is clarified in the revised manuscript as also suggested by reviewer 1:

Lines 153-158: “As mentioned above, Y336N represented a reversion in the natural evolution of human H3N2 NA (e.g. reverting from Bil71 to Bil69, Figure S1D). In contrast, Y336H was the forward mutation in the natural evolution of human H3N2 NA (e.g. mutating from SD93 to Mos99, Figure S1C). Since both Y336N and Y336H enabled SD93 to mutate away from Tyr at residue 336, we also tested whether Y336H could act as a permissive mutation for N387K.”

Lines 162-164: “This observation shows that the combination of E248G/R249K/Y336H/K344E/E369K potentiates the ability of K385N to compensate the fitness defect of N387K.”

Since the goal of the experiment was to search for permissive mutations for N387K, it was sufficient to draw the conclusion by only testing the mutations in the presence of N387K.

Nevertheless, we agree that additional mechanisms exist, which warrants future studies to explore. In fact, our discussion section already states that (see line 260 in the first submitted manuscript, which correspond to lines 287-288 in the revised manuscript):

“[The] biophysical constraints of NA evolution and antigenic drift remain to be fully comprehended.”

We are currently performing a follow-up study to identify additional mechanisms of epistasis in the natural evolution of NA.

The examination of the structures is cursory and limited to the more complicated “evolutionary network” related to N387K. If I interpreted the sequence alignment and presented structural panels correctly it would appear that additional substitutions in the strand harboring N387 should contribute to its mobility. This is not discussed. The contributions of the entirety of “mutant 8” mutations that recovered most of its infectivity is not discussed. A structural explanation for evolutionary contingency for N336Y is not discussed.

Response: As mentioned in the above response, we agree that additional mechanisms exist and are currently performing a follow-up study on them. Of note, this is the first study that characterizes the mechanism of epistasis in NA. While this study successfully elucidated a previously unknown mechanism of epistasis in NA, we believe that comprehending all the different mechanisms of epistasis in NA is outside the scope of this study.

That being said, in the first submitted manuscript, we have already acknowledged that the epistasis between N336Y and N356D is different from that between K385N and N387K (see lines 212-217 in the first submitted manuscript, which correspond to lines 231-236 in the revised manuscript):

“In contrast to N387K, N336H did not reduce the T_m of Bil69 NA. However, the first differential curve of the thermal shift assay for Bil69 N336H had an atypical shape (**Figure S5**). Similarly, the first differential curve for Bil69 N336Y showed two peaks. These results indicate the presence of multiple oligomeric states, hence aggregations²⁰. The shapes of their first differential curves returned to WT-like when N356D was introduced. As a result, the biophysical basis of epistasis between K385N and N387K is different from that between N336H/Y and N356D.”

Also, while we are not confident in providing a structural explanation for the more complicated “evolutionary network” related to N387K, some speculation is provided in the revised manuscript (lines 280-283):

“Nevertheless, while K385N can compensate the destabilizing effect of N387K in both SD93 and Mos99, it could only compensate the fitness defect of N387K in Mos99 but not SD93. This observation indicates that other permissive mutations for N387K, such as E248G, R249K, Y336H, K344E, and E369K, involve a stability-independent mechanism.”

Minor:

Throughout the manuscript graphs are presented with variable axes. Comparisons between panels are difficult. For instance: results of the mutational scanning are plotted quite differently, infectivity values differ by multiple orders of magnitude across panels of the same figure (e.g.

Figures 1 or 4) and the melting temperatures of various HAs differ by as much as 10 degrees in side-by side comparisons.

Response: This comment is a little bit confusing.

For example, we are unclear what is meant by: “infectivity values differ by multiple orders of magnitude across panels of the same figure (e.g. Figures 1 or 4)”. In Figure 1, infectivity values of our positive controls (i.e. WT) are quite consistent. In Figure 4, none of the panels concern about infectivity values.

In addition, this criticism is unfounded: “melting temperatures of various HAs differ by as much as 10 degrees in side-by side comparisons.” Firstly, our study focuses on NA (neuraminidase) instead of HA (hemagglutinin). Secondly, our unpublished data indeed show that the melting temperatures of various NAs from different strains indeed vary quite substantially (see Figure R1 below). Of note, the results are highly reproducible across three independent replicates. Given the substantial evolution of NA over time, it is not surprising to see that the NA melting temperature varies from strain to strain. In fact, this point was already discussed in our first submitted manuscript (see lines 257-259 in the first submitted manuscript, which correspond to lines 285-287 in the revised manuscript):

“Furthermore, wild type strains with very different NA stability can have similar virus replication fitness (Figure 1C and Figure 6A). Together, our results suggest that protein stability is only one of the biophysical determinants of NA for virus replication fitness.”

Moreover, a previous study on influenza nucleoprotein (NP) protein also demonstrates a similar phenomenon of melting temperature fluctuation during the course of virus evolution (see Figure 7C in PMID: 23682315).

We are currently in the process of understanding the mechanistic relationship between the natural evolution of NA and melting temperature. We hope to publish such result in a separate study.

[REDACTED]

As for the difference in the plots of the two mutational scans, it is mainly due to the de novo mutation in the first scan. “Enrichment” is typically used as a proxy for mutational fitness in a mutational scanning experiment. To compute the “enrichment” of a given mutant, its frequency information in the input library is needed. However, because a de novo mutation is not present in the input library of the first mutational scan, “enrichment” cannot be used to quantify the fitness of mutants that carries a de novo mutation. This is clarified in the revised manuscript (lines 139-145):

“Of note, Y336N was not included in the construction of the mutant library, and hence was a de novo mutation that emerged during viral passaging. Interestingly, Y336N represented a reversion to the ancestral stains, including HK68 and Bil69 (**Figure S1A and S1D**). While the fitness of each mutant in a mutant library could typically be estimated from its frequency change between the input mutant library and the post-passaged mutant library (i.e. frequency enrichment)¹⁶, the absence of Y336N in the input mutant library prevented the use of frequency enrichment for fitness measurement.”

Reviewers' comments:

Reviewer #2 (Remarks to the Author):

Lei et al., have submitted a revised manuscript that details evolutionary contingency in the evolution of influenza neuraminidase. Many of the revisions were to address clarity within the text and to include work with an authentic virus. Overall the authors define epistatic relationships that influence the identity of positions 336 and 387. These and other networks may govern possible antibody escape pathways.

Some reservations are unchanged and I apologize that they were not clearly articulated in my initial comments.

It is clear that mutations at positions 336 and 387 are context dependent. A network of mutations is required to restore fitness to N387K in the SD93 background. My question is what is the consequence of acquiring these two mutations? Do they profoundly alter the antigenic characteristics of NA, its enzymatic activity, or enable other mutations to occur? The authors should be commended on identifying these networks but at current the manuscript is missing a mechanism/explanation for their consequences.

A key question is the phenotype of the SD93 mutant combinations in the absence of N387K. Are these mutant combinations fitness neutral and permit N387K to occur or does the acquisition of N387K provide a benefit to the virus?

To clarify one of my previous comments, The Y-axis on a number of graphs are inconsistent. For example, the TM data in Figure 6A (different starting integers, ranges and top integers) or viral titers in figures 1C and D (8 log 10 units vs 6). A common Y-axis would make it easier for the reader to compare panels.

Minor.

It would be helpful to discuss that two sites in 344 and 369 changed shortly after circulating viruses acquired N387K. Can the authors speculate on how these mutations are tolerated in the N387K background?

Reviewer #2 (Remarks to the Author):

Lei et al., have submitted a revised manuscript that details evolutionary contingency in the evolution of influenza neuraminidase. Many of the revisions were to address clarity within the text and to include work with an authentic virus. Overall the authors define epistatic relationships that influence the identity of positions 336 and 387. These and other networks may govern possible antibody escape pathways.

Some reservations are unchanged and I apologize that they were not clearly articulated in my initial comments.

Response: We apologize for the misunderstanding during our previous revision. The remaining concerns are now addressed in the revised manuscript.

It is clear that mutations at positions 336 and 387 are context dependent. A network of mutations is required to restore fitness to N387K in the SD93 background. My question is what is the consequence of acquiring these two mutations? Do they profoundly alter the antigenic characteristics of NA, its enzymatic activity, or enable other mutations to occur? The authors should be commended on identifying these networks but at current the manuscript is missing a mechanism/explanation for their consequences.

Response: Thank you for the questions. In fact, residue 336 belongs to a classic antigenic region, suggesting its importance in antigenic drift. This is now clarified in the revised manuscript.

See lines 268-269: "Several residues in the epistatic network are within or immediately adjacent to known antigenic regions. For example, residues 336 and 369 are at antigenic regions I and III^{24,25}, respectively, whereas residues 248 and 249 are immediately adjacent to a recently identified epitope²⁶."

In contrast, to the best of our knowledge, functional consequences of the mutations at residue 387 have not been reported in the literature. [Redacted]. Since the conformation of this surface loop is reshaped by mutation N387K, along with K385N (Figure 6D-E in the revised manuscript), it is likely that N387K plays a role in the antigenic drift of NA.

[Redacted]

Furthermore, we speculate that N387K facilitated the emergence of K369T, which is within a known antigenic region (see response to the minor comment below).

A key question is the phenotype of the SD93 mutant combinations in the absence of N387K. Are these mutant combinations fitness neutral and permit N387K to occur or does the acquisition of N387K provide a benefit to the virus?

Response: The rescue titers of SD93 mutant combinations in the absence of N387K are reported in the revised manuscript as Figure S3B.

See lines 151-152: "Of note, in the absence of N387K, these mutation combinations had a higher virus replication fitness than the WT (Figure S3B)."

To clarify one of my previous comments, The Y-axis on a number of graphs are inconsistent. For example, the TM data in Figure 6A (different starting integers, ranges and top integers) or viral titers in figures 1C and D (8 log₁₀ units vs 6). A common Y-axis would make it easier for the reader to compare panels.

Response: Thank you for the clarification. The y-axis of Figure 6A, as well as Figure 1C and 1D are now modified to improve consistency.

Minor.

It would be helpful to discuss that two sites in 344 and 369 changed shortly after circulating viruses acquired N387K. Can the authors speculate on how these mutations are tolerated in the N387K background?

Response: After acquiring N387K, mutation at residue 344 (E344K) did not emerge after a decade later. Therefore, we do not think the emergence of mutation at 344 is influenced by N387K. However, mutation at residue 369 (K369T), which emerged within a few years after N387K arose, is possibly promoted by N387K. The potential epistatic relationship between K369T and N387K is described in the revised manuscript.

See lines 262-265: "We also noticed that mutation K369T emerged in 2009-2010 influenza season, which was within 2 years after the fixation of N387K. N387K added a positively charged residue on the NA surface, whereas K369T did the opposite. As a result, it is possible that N387K promoted the emergence of K369T due to charge balancing⁹."